# Pre-Training on Mixed Data for Low-Resource Neural Machine Translation

**Wenbo Zhang** [1,2,3] **, Xiao Li** [1,2,3,*] **, Yating Yang** [1,2,3,*] **and Rui Dong** [1,2,3]

1. The Xinjiang Technical Institute of Physics & Chemistry, Chinese Academy of Sciences, Urumqi 830011, China; zhangwenbo16@mails.ucas.edu.cn (W.Z.); dongrui@ms.xjb.ac.cn (R.D.)
2. University of Chinese Academy of Sciences, Beijing 100049, China
3. Xinjiang Laboratory of Minority Speech and Language Information Processing, Urumqi 830011, China
* Correspondence: xiaoli@ms.xjb.ac.cn (X.L.); yangyt@ms.xjb.ac.cn (Y.Y.)

**Abstract:** The pre-training fine-tuning mode has been shown to be effective for low resource neural machine translation. In this mode, pre-training models trained on monolingual data are used to initiate translation models to transfer knowledge from monolingual data into translation models. In recent years, pre-training models usually take sentences with randomly masked words as input, and are trained by predicting these masked words based on unmasked words. In this paper, we propose a new pre-training method that still predicts masked words, but randomly replaces some of the unmasked words in the input with their translation words in another language. The translation words are from bilingual data, so that the data for pre-training contains both monolingual data and bilingual data. We conduct experiments on Uyghur-Chinese corpus to evaluate our method. The experimental results show that our method can make the pre-training model have a better generalization ability and help the translation model to achieve better performance. Through a word translation task, we also demonstrate that our method enables the embedding of the translation model to acquire more alignment knowledge.

**Keywords:** neural machine translation; pre-training; low resource; word translation





## 1. Introduction

In recent years, neural machine translation (NMT) has achieved rapid development [1–3]. End-to-end neural machine translation is increasingly being embraced by researchers [4–7]. An NMT model is usually based on the encoder-decoder architecture. In early models of NMT, the encoder converts a variable length source language sentence into a fixed-length context vector, then the decoder generates target language words one by one from the fixed context vector [4]. After the emergence of the attention mechanism [5,6], the output of the encoder is no longer a fixed-length context vector, but multiple context vectors of the same length as the input, and the decoder generates target language words according to the variable context vector which is a weighted sum of the multiple context vectors. NMT has reached the level of statistical machine translation (SMT). With the introduction of the NMT model based on convolutional neural networks and the transformer [2,7] model, NMT has surpassed SMT as the most popular method. Some studies even claim that their NMT system has achieved human parity in some domains for some languages [3].

Due to rapid advances in NMT, we have witnessed dramatic improvements in the quality of machine translation. However, these achievements are mostly based on a large parallel corpus. When language pairs and domains change, it is usually difficult to obtain parallel corpora large enough to train a high-quality NMT model. Although NMT has achieved impressive progress in resource-rich language pairs, it has a poor performance when there are few parallel corpora. Therefore, how to implement a high-quality neural machine translation system on a small corpus has become an urgent problem to be solved.

Zoph et al. [8] and Nguyen et al. [9] try to use transfer learning to transfer knowledge from high-resource language pairs to low-resource language pairs, but this method may not be applicable for non-similar languages. Parallel corpus are expensive and difficult to obtain. On the contrary, monolingual data are widely available on the Internet and are easy to obtain. A large number of methods try exploit monolingual data to improve low-resource neural machine translation [10–15].

Recently, pre-training methods based on the transformer structure have shown great benefits in the field of natural language processing [16,17]. Their biggest benefit is that these methods use a large amount of monolingual data to pre-train, and then fine-tune pre-trained models on a small amount of task-specific data. Lample and Conneau [18] apply this pattern to a neural machine translation task through a method called a cross-lingual language model (XLM). Similarly, Song et al. [19] propose a masked sequence to sequence a pre-training model (MASS) to enhance NMT through monolingual data. The two methods first randomly mask a fragment of the sentences on both source-side and target-side monolingual data, and then train the pre-training models by predicting these masked fragments given the unmasked fragments. Using pre-trained models to initialize NMT models can greatly improve the translation quality.

In this paper, we propose a new method that builds on XLM and MASS to achieve better results in low-resource neural machine translation tasks. The pre-training models XLM and MASS learn knowledge from monolingual data by predicting the masked words according to unmasked words. On the basis of the original pre-training method, our method randomly replaces some unmasked words with their translation words in another language. In our method, the pre-training model also predicts masked words based on unmasked words. Unlike the original pre-training method, the unmasked words in our method may contain words from both the source and target languages. It should be noted that our method requires a bilingual lexicon, which can be extracted from parallel corpora using the word alignment technique of SMT. On the one hand, our method uses both monolingual and bilingual data, and on the other hand, a sentence used for pre-training may contain the words in both the source and target language. Therefore, our pre-training model is trained on mixed data. We conduct experiments on Uyghur-to-Chinese and Chinese-to-Uyghur translation tasks, and our method achieves improvements on both these tasks. The experimental results also show our method can alleviate the over-fitting of pre-training models. Last but not least, we propose a simple word translation model through which we demonstrate that our method can help the embedding of the translation model to acquire more alignment knowledge. In summary, the contribution of this paper are list as follows:

1. We propose a new pre-training method, which can exploit both monolingual and bilingual data to train the pre-training model;
2. We conduct experiments on Uyghur-Chinese corpus, which is a realistic low resource scenario. Experimental results show that the pre-training model in our method has a better generalization ability and improves the performance of the translation model;
3. We propose a word translation model to measure the alignment knowledge contained in the embedding of other models;
4. The word translation model mentioned above proves that our method allows translation models to learn more alignment knowledge.

The layout of this paper is as follows: In Section 2 we discuss the related works; in Section 3 we introduce our pre-training method; in Section 4 we present the word translation model and explain how it can measure the alignment knowledge in other models; in Section 5 we describe the details of our experiments; and finally, the conclusion is drawn in Section 6.

## 2. Related Works

In the era of SMT, a language model plays an important role in the translation system, which is considered to be able to boost fluency of translations [20]. Gulcehre et al. [10] inte-

grate a recurrent neural network (RNN) language model pre-trained on target monolingual data into NMT, and predict the target next word by scoring it using both the NMT decoder and the language model. Back-translation [11] uses target-side monolingual data and a reverse translation model to generate additional synthetic parallel data, which can also be traced back to SMT and has been proven to be very effective in NMT. All of these methods utilize target-side monolingual data to improve NMT. Zhang and Zong [12] propose to use a self-learning algorithm and multi-task learning framework to exploit source-side monolingual data. Xia et al. [13] propose a dual learning method based on back translation, which uses an NMT model to translate monolingual sentences forward to the other language and then reconstruct it by another reverse translation model. This method can utilize both source and target monolingual data through two reverse translation models, but the optimization of this method is too complex to be practical.

The popularity of pre-training models has led many researchers to use pre-training models to enhance NMT. For example, Skorokhodov et al. [21] use the source language model and target language model trained on monolingual data to initialize the encoder and decoder of the translation model respectively. With the emergence of the pre-training model BERT [17], researchers have realized that the pre-training model based on transformer has great potential to transfer the knowledge of monolingual data to other models. Several approaches have been proposed to improve neural machine translation using BERT or a pre-training model similar to BERT. Zhu et al. [22] propose a BERT-fused model which uses attention mechanisms to fuse representations of the NMT model with that of BERT. Guo et al. [23] take two different BERT models as the encoder and decoder respectively and use a lightweight module called an adapter to connect the encoder and decoder. However, all of these approaches need to change the NMT model to integrate with BERT. XLM [19] and MASS [20] are both parameter initialization methods, and their pre-training models are pre-trained on monolingual data in both source and target languages. XLM uses the encoder of the transformer to pre-train and initializes the encoder and decoder of the NMT model with the pre-trained model. The difference is that the pre-training model of MASS is based on the overall NMT model. Another difference is that XLM randomly masks the words of a sentence for pre-training, while MASS randomly masks a continuous segment of a sentence.

## 3. Our Method

Our method is based on XLM [19] and MASS [20], so let us first cover the details of these two approaches. See Figures 1a and 2a for the illustrations of XLM and MASS, respectively. XLM alternately trains the masked language modeling (MLM) [17] on the source and target languages monolingual data. Specifically, for each input, XLM samples randomly 15% of the input tokens, replaces them by a specific symbol [MASK] 80% of the time, by a random token 10% of the time, and keep them unchanged 10% of the time. MASS also alternately uses both source and target monolingual data to train the pre-training model. The difference is that for each input, MASS randomly masks a contiguous fragment that makes up 50% of the input. Both XLM and MASS predict the masked tokens by an unmasked context. Another difference is that the XLM uses an encoder to predict the masked tokens and MASS uses the decoder to do that. MASS predicts the masked fragment based on the unmasked context, like an NMT model, in which the unmasked context is the source sentence and the masked fragment is the target sentence. More details are available in XLM and MASS.

The core idea of our method is replacing unmasked words with its translation words randomly after XLM or MASS has masked the input. The illustrations are shown in Figures 1b and 2b. Specifically, for each masked input, we randomly replace each unmasked word with its translation words in another language 50% of the time, and do not modify the other 50% of the time. In this way, the pre-training model can indirectly realize that in an unmasked context the replaced tokens and its translation word have the same semantics because for the pre-training model, the sentence in which the unmasked

context is processed by the replacement operation and the sentence in which the unmasked context is unchanged have the same output. It should be noted that we do not replace all the unmasked words with its translation words, but for each unmasked word, we first check see if there is a translation word for the word in the bilingual lexicon, and replace it with 80% probability if there is. The masking mechanism of XLM and MASS is a data enhancement approach because a sentence can be randomly masked to produce different input. Our method can generate more and different inputs based on XLM and MASS, so our method is a better data enhancement approach and the over-fitting of pre-training model can be mitigated to some extent.

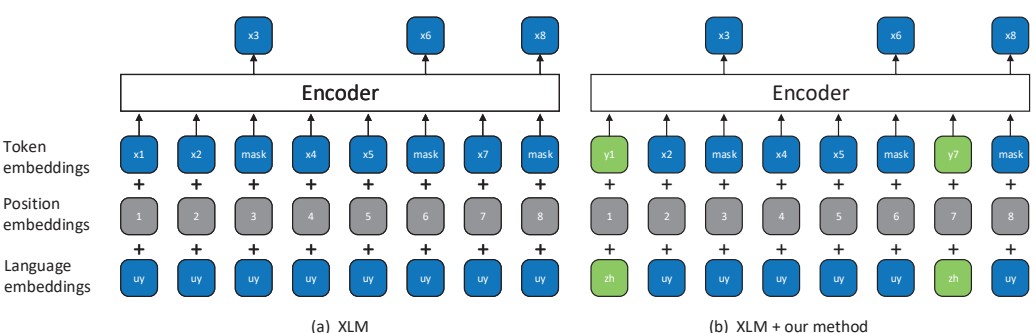

**Figure 1.** The examples of the cross-lingual language model (XLM) and our method based on XLM. (**a**) XLM randomly masks some tokens and then predicts the masked tokens by unmasked tokens and (**b**) our method based on XLM replaces some unmasked tokens with its translation tokens.

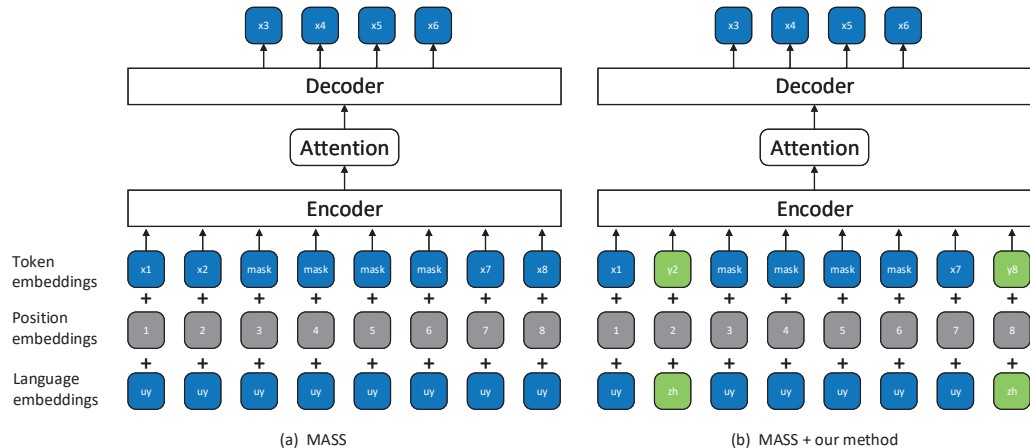

**Figure 2.** The examples of masked sequence to sequence (MASS) and our method based on MASS. (**a**) MASS randomly masks contiguous tokens and then predicts the masked tokens by unmasked tokens and (**b**) our method based on MASS replaces some unmasked tokens with its translation tokens.

To obtain the translation words of unmasked words, we extract a bilingual lexicon from a parallel corpus. A parallel corpus is composed of sentence pairs that contain a source language sentence and a target language sentence, and the two sentences express the same meaning. IBM model 2 [24], which is often used as part of SMT, can align source language words with target language words. The detailed extracting of bilingual lexicon has three steps: (1) use fast_align [25] which is a simplified variant and fast implement of IBM model 2 to align source words with target words in parallel corpus; (2) filter out word pairs that occur less than three times because they may be due to incorrect alignment; and (3) remove word pairs that contain more than three translation words for one word. We think these word pairs are mainly caused by function words or polysemous words and replacing polysemous words requires context, so we remove these cases as well. These steps can enable an accurate bilingual lexicon from a parallel corpus. For an unmasked

word, we find all its translation words in the bilingual lexicon, and choose one of them to replace it randomly.

## 4. Word Translation Model

To measure the alignment information of different models (such as the pre-training model and the NMT model), we proposed a word translation model, whose architecture is shown in Figure 3. The input of the word translation model is a word, the output is its translation word in another language. The word translation model is composed of four layers of hidden units. The first layer is the embedding layer. The second and third layers are the fully connected layers. The fourth layer is the classification layer and a standard softmax function. In addition, the embedding and classification layers share parameters that are initialized by other models and are frozen during training. Therefore, the parameters of the word translation model only contain the parameters of two fully connected layers, and the word translation model is formulated as:

$$v_t = W^2 ReLU(W^1 v_s + b^1) + b^2. \tag{1}$$

where $ReLU(x) = max(0, x)$ is the activation function, $W^1, b^1$ and $W^2, b^2$ are the weight matrix and bias parameters for layer 2 and layer 3 respectively. $v_s$ and $v_t$ are word vectors in both languages from the embedding layer.

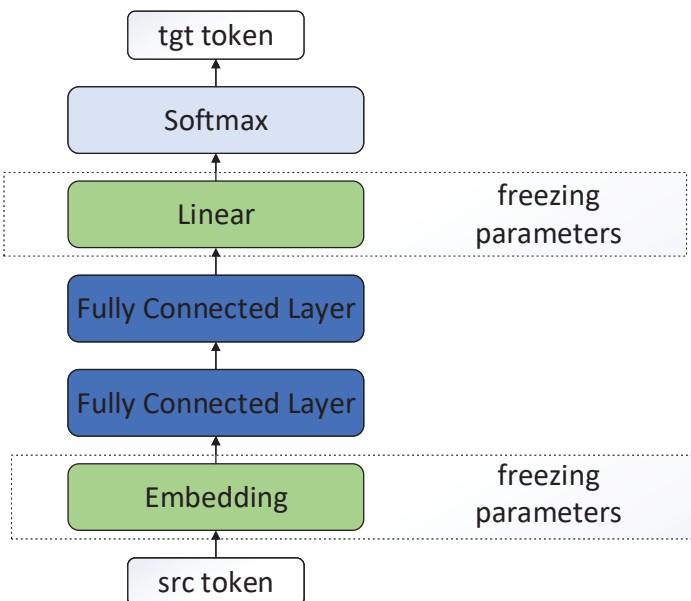

**Figure 3.** The model architecture of word translation model. The embedding layer and linear layer are initialized by the cross-lingual word embedding, and their parameters are frozen on the word translation task.

It is obvious that the word translation model is a very simple model. From the perspective of model structure, the structures that affect the quality of word translation include the word embedding layer and full connection layer. In our experiments, the size of the word dictionary is 60,332, the dimension of the word vector is 512, and the dimensions of the two fully connected layer matrices are 512 × 1024 and 1024 × 512 respectively. Therefore, the number of embedding parameters is 30,889,984, and the number of fully connected laye parameters is 1,068,544. It can be seen that the number of parameters in the word embedding layer is 30 times that of the full connection layer, so the influence of word embedding layer on the quality of word translation is more important than that of the full connection layer. In addition, the parameters of the word embedding layer are all from other models. We can conclude that the quality of word embedding layer in different

models is the main factor for word translation results and we can measure the alignment information in the embedding of different models by comparing the performance of the word translation models initialized by the different models. In fact, in the pre-training model and the neural machine translation model, the number of parameters of the word embedding layer also accounts for a very large proportion in the whole model. For example, in the pre-training model XLM, the number of parameters of the word embedding layer accounts for more than half of the proportion. To some extent, the amount of alignment information in the word embedding layer represents the amount of alignment information in the model.

## 5. Experiments and Results

### 5.1. Datasets and Preprocessing

We conduct experiments on Uyghur-Chinese dataset, which is a realistic low resource scenario. In our experiments, both parallel corpus and monolingual data are used to build different neural machine translation systems. The parallel corpus from the evaluation conference CCMT2019 consists of 170,061 training sentence pairs, 1000 validation sentence pairs, and 1000 test sentence pairs. The CCMT2019 also provides Chinese monolingual data containing 7.42 million Chinese sentences. We collected 3 million Uyghur sentences from Uyghur news websites as Uyghur monolingual corpus. Many studies have shown that domain relevance between monolingual data and parallel corpus has a great influence on translation results. So we filtered monolingual data based on word coverage. For Chinese monolingual data, we filtered sentences in which more than 6% of the words do not appear in the Chinese portion of parallel corpus. For Uyghur monolingual data, we filtered sentences in which more than 10% of the words do not appear in the Uyghur portion of parallel corpus. As a result, we indirectly maintain the domain correlation between the source-side monolingual data and target-side monolingual data by maintaining the high domain correlation of both sides of the monolingual data and parallel corpus. Table 1 shows the statistics of the used data.

**Table 1.** Preprocessed parallel and monolingual data.

| Dataset | Training Size | Validation Size | Test Size |
| --- | --- | --- | --- |
| Uyghur-Chinese para | 0.17M | 1k | 1k |
| Uyghur mono | 2.45M | - | - |
| Chinese mono | 2.62M | - | - |

We used Moses [26] script (https://github.com/moses-smt/mosesdecoder accessed on 18 March 2021) to tokenize the Uyghur data and Jieba tool (https://github.com/fxsjy/jieba accessed on 18 March 2021) for Chinese word segmentation. We performed byte pair encoding (BPE) [27] on joint vocabulary with 50,000 merge operations, and the corresponding vocabulary size in the NMT model is 60,332.

### 5.2. Systems and Model Configurations

We consider different pre-training models and then use these pre-training models to initialize the same translation model for fine-tuning on parallel data. In this paper, we compare the results of translation models initialized by the following five pre-training models.

- No pre-training: All parameters in the translation model are initialized randomly as in the traditional translation model;
- XLM [19]: We train the pre-training model through masked language modeling as described in XLM;
- XLM + our method: On the basis of XLM, we train the pre-training model by further replacing the unmasked words according to our method mentioned above;
- MASS [20]: We train the pre-training model through masked sequence to sequence modeling as described in MASS;

- MASS + our method: On the basis of MASS, we train the pre-training model by further replacing the unmasked words according to our method mentioned above.

We used the transformer-base proposed in the paper as the basic model structure, which means we used 6 layers for both the encoder and decoder, set the dimension of word embedding as 512, and the head number as 8. We used the Adam [28] optimizer for all models. All models are trained on two 32G Tesla V100 GPUs. For all pre-training models, the dropout [29] is set as 0.1 and the learning rate is set as 0.0001. For the NMT model, we set the dropout rate to 0.3 and the initial learning rate to 0.0005.

*5.3. Results*

During inference, we use beam search with a beam size of 4 for the NMT model. We comput the BLEU [30] score with multi-bleu.pl (https://github.com/moses-smt/mosesdecoder/blob/master/scripts/generic/multi-bleu.perl accessed on 18 March 2021) as the evaluation metric. Table 2 shows the experimental results of NMT models initialized by different pre-training models.

**Table 2.** The results of different translation systems on test sets.

| Training Data | System | uy-zh | zh-uy |
|---|---|---|---|
| bilingual data | no pre-training | 27.50 | 22.28 |
| bilingual + monolingual data | XLM | 31.31 | 24.80 |
| | XLM + our method | 31.83 | 25.51 |
| | MASS | 31.34 | 24.58 |
| | MASS + our method | 31.61 | 25.27 |

For Uyghur-to-Chinese and Chinese-to-Uyghur machine translation tasks, from Table 2, we can see that all pre-trained translation models perform much better than the not pre-trained translation models. Furthermore, our method performs better on the basis of both XLM and MASS models for all translation tasks. More specifically, our proposed method based on XLM achieves 0.52 and 0.71 BLEU points improvements compared to XLM on Uyghur-to-Chinese and Chinese-to-Uyghur translation directions respectively. Compared to the MASS baseline, our proposed method based on MASS also achieves 0.27 BLEU points improvement on the Uyghur-to-Chinese translation direction and 0.69 BLEU points improvement on Chinese-to-Uyghur translation direction.

We report the perplexity (PPL) scores of all pre-training models in Figure 4. Our method has a lower PPL score on the validation set when the training time is long enough. This is evident from Figure 4a,b,d. These results indicate that our method can alleviate the over-fitting of the pre-training models.

*5.4. Analysis*

Our method uses bilingual lexicons from bilingual data to improve pre-training models. Can the pre-training models learn these bilingual lexicons knowledge through the proposed method? Can our method help the NMT model learn more alignment knowledge? We study these problems through the word translation task proposed above.

We take the bilingual lexicons for pre-training in our method as the dataset of word translation task, and divide it into a training set, validation set, and test set. The statistics of dividing are shown in Table 3.

**Table 3.** Word translation dataset.

| Total Num | Training Num | Validation Num | Test Num |
|---|---|---|---|
| 35,159 | 30,000 | 1159 | 4000 |

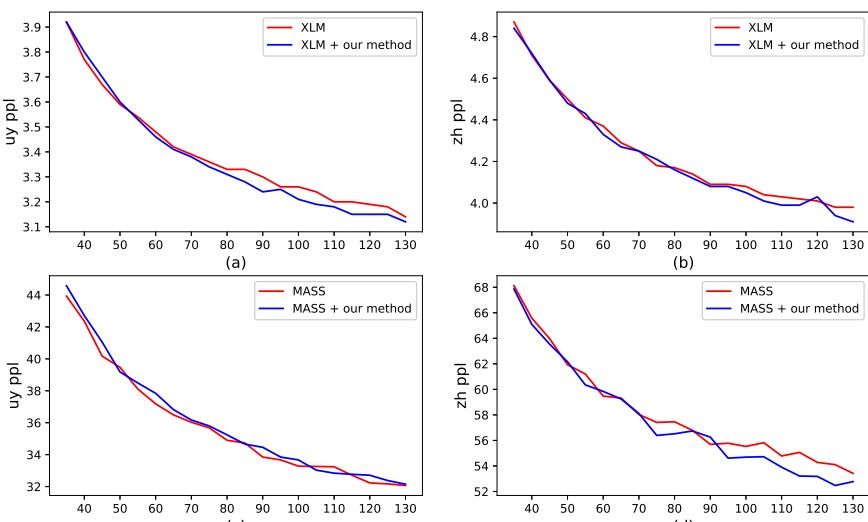

**Figure 4.** The perplexity (PPL) scores of different pre-training models on the validation set with respect to the epoch during pre-training. (**a**) the Uyghur results of XLM and our method based on XLM; (**b**) the Chinese results of XLM and our method based on XLM; (**c**) the Uyghur results of MASS and our method based on MASS; and (**d**) the Chinese results of MASS and our method based on MASS.

We use the embedding of all pre-training models to initiate the word translation model proposed above and then use a stochastic gradient descent (SGD) optimizer with a learning rate learning rate of 0.15 to optimize the word translation model. The batch size is set to 512 and dropout is set to 0.1. We also try to use fastText (https://github.com/facebookresearch/fastText accessed on 18 March 2021) to train a traditional word embedding model to initialize the word translation model for comparison. We take the PPL score as the metric of word translation task. The results are reported in Table 4.

**Table 4.** The PPL scores of word translation models initialized by different embeddings on each test set.

| Embedding | fastText | XLM | XLM + Our Method | MASS | MASS + Our Method |
|:---:|:---:|:---:|:---:|:---:|:---:|
| **PPL** | 19610 | 31.57 | 5.88 | 52.93 | 10.67 |

According to Table 4, it can be seen that the embedding of the pre-training model has a better PPL score than that of the traditional word embedding model. This suggests that the embedding of the pre-training model is very suitable for the translation model to learn the alignment knowledge between source and target language. We can also find that our method performs better than the original pre-training model, which means that our method can make the pre-training model learn this alignment knowledge by replacing the unmasked words with its translation words.

Similarly, we compare the embedding of NMT models initialized by different pre-training models in the same way. The results are presented in Table 5. All the NMT models are on Uyghur-to-Chinese translation direction from Table 2. As shown in Table 5, pre-trained NMT models have a better performance than that of the no pre-trained NMT model. Our method has a lower PPL score than that of the original pre-training models. In combination with Tables 4 and 5, it can be seen that after fine-tuning on the parallel corpus, the embedding of the model is improved in the word translation task. Our method can further improve the performance of models in word translation task. In other words,

the pre-training model can make it easier for the NMT model to learn alignment knowledge in parallel corpus, and our method further enhances this ability.

**Table 5.** The word translation results of neural machine translation (NMT) models initialized by different pre-training models on each test set.

| NMT | No Pre-Training | XLM | XLM + Our Method | MASS | MASS + Our Method |
|---|---|---|---|---|---|
| PPL | 52.80 | 14.63 | 4.99 | 25.11 | 8.23 |

## 6. Conclusions

In this paper, we proposed a simple method to improve the performance of the pre-training approaches, which are based on a masking mechanism and predict the masked tokens using unmasked context. After the input tokens masked randomly by pre-training approaches such as XLM or MASS, our method randomly replaces part of the unmasked context with its translation tokens according to a bilingual lexicon extracted from parallel corpus. Experimental results on Uyghur-Chinese dataset showed that our method could alleviate the over-fitting of the pre-training model and achieve better performance in both the pre-training and translation stages. In addition, we also designed a word translation model to measure alignment knowledge in different models by initializing the word translation model with the embedding layer of different models. The results showed that our method could make the pre-training model learn more alignment knowledge, so that the translation model initialized by the pre-training model could learn alignment knowledge more easily.

**Author Contributions:** Conceptualization, W.Z.; investigation, W.Z.; methodology, R.D., X.L.; software, W.Z.; validation, Y.Y., R.D. and W.Z.; funding acquisition, X.L., Y.Y. All authors have read and agreed to the published version of the manuscript.

**Funding:** This work is supported in part by the Subsidy of the Youth Innovation Promotion Association of the Chinese Academy of Sciences (2017472), the Xinjiang High-level Talent Introduction Project (Xinrenshehan [2017] No. 699), The Western Light of the Chinese Academy of Sciences under Grant No.2017-XBQNXZ-A-005, and the National Key Research and Development Program of China (2018YFC0823404).

**Acknowledgments:** The authors would like to thank the editor in chief and all anonymous reviewers for their constructive advice.

**Conflicts of Interest:** The authors declare no conflict of interest.

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
