# Peer review of "Pre-Training on Mixed Data for Low-Resource Neural Machine Translation"

_information, doi:10.3390/info12030133_

Round 1

Reviewer 1 Report

The paper proposes a simple pre-training method that goes beyond the use of monolingual data with masked words to use of word translations instead of MASK tokens. The authors show nice improvements on a low-resource language pair.

This is a small but effective research contribution. I am not familiar with the standards of the journal but this paper passes the threshold of being accepted in the highly competitive conferences of the field (ACL, EMNLP, etc.).

The paper is clearly written and only suffers occasionally from fluency issues. Some are noted below.

Line 28: "fields" -> "data conditions" or "domains for some languages"

Line 31: "large-scale parallel corpus" -> "large parallel corpora" 

Line 32: "enough parallel corpus" -> "parallel corpora large enough"

Line 32: "an advanced" -> "a high-quality" (also elsewhere)

Line 35: "small scale" -> "small"

Line 39: "corpus is" -> "corpora are"

Line 37: "advantages" -> "benefits" (also elsewhere)

Line 66: "fragment" -> "fragments"

Line 68: "collaborates with" -> "builds on"

Line 83: "over fitting" -> "over-fitting" or "overfitting"

Line 151: "simple" -> "simplified variant"

Line 153: can you clarify what is happening here? it sounds like that you throw out translations for frequent pairs that tend to have a longer tail of paired translations.

Line 230: "no pre-trained" -> "not pre-trained"

Author Response

Thank you for your affirmation and suggestions. Please refer to the attachment for specific modifications.

Reviewer 2 Report

A paragraph to be included describing the layout of the paper

The "Related Work" section is very short relative to the introduction. I suggest to add some material  from the introduction here to make it shorter the "introduction" and enlarge the related work part.

The methodology section writing is hard to be read.

I didn't understand some sentences like: "In this paper, we propose a new method that can cooperate with the mask-based pre-training approaches. Our method first extracts a bilingual lexicon from parallel corpus and then randomly replaces unmasked words in monolingual data" lines 274 -276

Contributions of the paper are not clear. For example:

  • I couldn't understand how the proposed method alleviated the over fitting??
  • couldn't understand  how the word translation model was used to measure the alignment knowledge in different models?

Author Response

Thank you for your suggestion, which makes this paper easier to understand. Please refer to the attachment for specific modifications.

Round 2

Reviewer 2 Report

The authors updated the paper taking into considerations my comments. However, I still have a problem with the contributions as I didn't see evidence for enhancing the quality of NMT using the given approach except enhancing the PPL score of the word translation model result but not a sentence to sentence translation as most MT systems do. Could the authors clarify this point?

Round 3

Reviewer 2 Report

Table-2 answer my question but you should have highlighted this results in the conclusion and in your contribution. Not mentioning it in the contribution let me think you missed it as you emphasized more the improvement in PPL.